# Experimental Study of Deep Submersible Structure Defect Monitoring Based on Flexible Interdigital Transducer Surface Acoustic Wave Technology

**DOI:** 10.3390/s23031184

**Published:** 2023-01-20

**Authors:** Zhongjun Ding, Zhiliang Feng, Hongyu Li, Dejian Meng, Yi Zhang, Dewei Li

**Affiliations:** 1National Deep Sea Center, Qingdao 266237, China; 2College of Ocean Science and Engineering, Shandong University of Science and Technology, Qingdao 266590, China

**Keywords:** deep submersibles, structural defect monitoring, interdigital transducer, PVDF

## Abstract

In view of the shortage of structural defect monitoring methods for deep submersibles, numerical simulation and experimental research on underwater SAW propagation based on interdigital transducers are carried out in this paper. PVDF interdigital transducer (PVDF-IDT) has shown considerable potential in the application of structural health monitoring because of its micro size, soft material characteristics, and the characteristics of long-term bonding on the surface of the tested structure. In order to realize the application of IDT on submersible or underwater structures, it is necessary to understand the influence of underwater environment on IDTs with different structures. The underwater attenuation of IDT with 2–5 mm wavelength and the underwater attenuation of Lamb (A0 mode) wave on a 4 mm thick titanium alloy plate is obtained through COMSOL software simulation. The experimental verification shows that the simulation results match with the actual situation, which proves that COMSOL software can accurately calculate the acoustic attenuation of surface waves at the solid–liquid interface. At the same time, the underwater attenuation of IDT with different structures is very different, providing important design parameters for the underwater interdigital transducer. In this paper, it is found that the Lamb wave has significant advantages over the Rayleigh wave in the health monitoring of underwater thin plate structures.

## 1. Introduction

At present, people’s interest in structural defect monitoring of offshore structures has greatly increased [1]. For marine equipment working in complex marine environments, the fatigue failure of key support may lead to continuous failure of adjacent support structures due to overload [2], resulting in serious accidents. Especially for manned submersibles working in an extreme deep-sea environment, once the slight defects expand and lead to structural failure or even fracture, there will be a huge hidden danger threatening the safety of human life and property. Zhao et al. [3] reported the damage mechanism of buoyancy materials for submersibles, and Qin et al. [4] made statistics and analyses on structural defects of submersibles, proving that structural defects of submersibles are common. Therefore, the research on health monitoring of marine equipment is a work of practical engineering significance.

As things are at the moment, health monitoring technology has been applied in many fields. However, there are few research cases on deep-sea equipment. Yang Huawei et al. developed a structural health monitoring system using a strain gauge as sensor to monitor the stress and strain of the submersible pressure structure [1]. However, the health monitoring technology of ships and offshore platforms is relatively widely used. The Lech Murawski et al. [5] team used fiber grating sensors [6,7] with modern technology on ship models to monitor some ship structures. Bai [8] used PVDF film as a sensor to detect the dynamic stress and strain of offshore platforms. The sensors used by the above scholars can achieve rational effects, but there are some shortcomings, such as sensors being too complex or insensitive to the damage of marine equipment structures.

To solve the above problems, PVDF-IDT is selected as the sensing element of structural health monitoring in this paper, and the detection of underwater structural defects is realized by exciting surface acoustic waves which can directly detect structural defects. Compared with other surface wave generation methods, the greatest advantage of IDT lies in its microphysical size and the high efficiency of converting electrical signals into mechanical vibration [9]. IDT with PVDF film as the base material can also be stuck to curved structures for a long time. It has been reported that IDT is used for structural health monitoring [10,11]. PVDF-IDT has shown considerable advantages in deep-sea equipment such as deep submersibles and submarines, which have strict requirements on sensor volume and system. However, the propagation characteristics of surface acoustic waves excited by PVDF-IDT at the solid–liquid interface are unknown. In this paper, the underwater propagation characteristics of surface waves will be studied through numerical simulation and experimental verification, and interdigital transducers for underwater applications have been designed and manufactured.

## 2. Sound Structure Coupling Theory

### 2.1. The Governing Equations of Sound Wave Propagation in Solids and Liquids

According to Newton’s second law, the acoustic wave propagation equation in solid materials can be written as:(1)$$\nabla T+F=\rho \frac{\partial^{2}u}{\partial^{2}t}$$
where T is the stress tensor; F is for physical force; ρ is the density; u is the displacement of the particle.

The propagation equation of sound wave in liquid can be written as:(2)$$\frac{1}{\rho c^{2}}\frac{\partial^{2}p_{t}}{\partial t^{2}}+\nabla \cdot \left(-\frac{1}{\rho }\left(\nabla p_{t}-q_{d}\right)\right)=Q_{m}$$
where $p_{t}\;$is sound pressure; c is the velocity of sound; ρ is the fluid density; $q_{d}\;$is a monopole source; $Q_{m}$ is the dipole source.

### 2.2. Acoustic Structure Coupling Equation

In classical acoustics, the physical assumption for solving the Helmholtz equation or scalar wave equation is that the fluid is inviscid isentropic fluid. In this case, the slip condition is applied to the solid wall and the coupling is realized in the normal direction of the surface. Acousto-structure coupling is a multi-physical field phenomenon, where the sound pressure loads the solid, and the structural acceleration (normal acceleration of the solid wall) will continuously stimulate the fluid.

The motion equation of linear acoustics can be written as:(3)$$\rho \frac{\partial V}{\partial t}+\nabla p_{t}=0$$
where V is the velocity of fluid particle. As can be seen from Figure 1, solid excitation to fluid is normal acceleration, so Equation (3) can be transformed into:(4)$$-n\cdot \left(-\frac{1}{\rho }\nabla p_{t}\right)=-nu_{tt}$$
where: n is the normal vector of the solid wall and $u_{tt}\;$is the normal acceleration of the solid wall. Considering the influence of sound pressure on solid and other sound sources of fluid, the sound-structure coupling formula can be written as
(5)$$\left\{ \begin{array}{c} -n\cdot \left(-\frac{1}{\rho }\left(\nabla p_{t}-q_{d}\right)\right)=-n\cdot u_{tt}\\ \;F_{A}=P_{t}n \end{array}\right.$$
where $q_{d}\;$is a unipolar source and $F_{A}\;$is the load on the structure.

## 3. Design and Manufacture of IDT Sensor

The typical structure of IDT consists of three layers; the top and bottom electrode layers are separated by piezoelectric layers, and the distance between two adjacent fingers of the interdigital electrode is adjusted to match the specific wavelength. IDT mainly monitors structural defects by exciting two kinds of surface waves, namely the Rayleigh wave and Lamb wave. When the Rayleigh wave is excited, the electrode width and electrode spacing of the IDT electrode pattern selected in this paper are equal, and the excitation wavelength is 2–5 mm.

When the Lamb wave is excited, the IDT is designed according to the dispersion curve of the titanium alloy plate shown in Figure 1a,b, and the mode of the excited Lamb wave is selected. In this paper, the Lamb wave of A0 mode is excited on a 4 mm thick titanium alloy plate, and the slope l of line Y in Figure 1a is λ/d, where λ is the IDT wavelength and d is the thickness of titanium alloy plate. Given the slope of line Y l = 1.25 and d = 4 according to Figure 1a,b, the wavelength of IDT can be obtained λ = 5 mm; the excitation frequency is 500 kHz.

The manufacturing process of IDT includes screen printing, electric field-driven jet deposition, laser etching [12] and flexible PCB manufacturing. To meet the waterproof requirements, flexible PCB is selected to make IDT with 2, 3 and 4 mm wavelengths. IDT with 5 mm wavelength is made through screen printing and water-based polyurethane filling waterproof. The reason for choosing different processing technologies is that the sensitivity of flexible PCB-IDT with a 100 μm thick copper electrode to stimulate a 5 mm wavelength surface wave is too low. The sensitivity of copper electrodes with different thicknesses is analyzed in the literature [13]. Figure 1d shows the sticking model of flexible PCB fingers on the structure [14,15,16], and Figure 1e shows the structural diagram of IDT made by screen printing, The structure of the IDT made by screen printing is divided into four layers from top to bottom, which are polyurethane layers, conductive silver adhesive layers, PVDF films, and tested structures. Figure 1c shows the interdigital electrode parameters of IDT with a wavelength of 5 mm. The difference between the electrode parameters of IDT with different wavelengths is the width of the interdigital electrode and the spacing between adjacent electrodes, and the two parameters are one-fourth of the corresponding IDT wavelength (See Table 1 for the number of electrodes). Figure 1f shows a conceptual diagram of an IDT receiving circuit applied underwater. Finally, the piezoelectric layer of IDT is a 110 μm PVDF film.

## 4. Numerical Analysis

In this paper, COMSOL software is used to analyze the impact of underwater environment on IDT, and the acoustic structure coupling model of IDT is established under the two-dimensional space model, and Equation (5) is taken as the boundary condition of the acoustic structure coupling model. As shown in Figure 2a, two interdigital transducers are arranged on a titanium alloy plate (TC4) with a thickness of 4 (Lamb wave) or 10 (Rayleigh wave) mm in a way of receiving and transmitting separation. and IDT of excited surface waves was simulated. The excitation signal shown in Figure 1b is an 8-cycle sine wave signal. In this paper, specific parameters of the simulation for the following cases can be seen in Table 1:(1)Rayleigh wave simulation of 2 mm wavelength in 10 mm thick titanium alloy plate;(2)Rayleigh wave simulation of 3 mm wavelength in 10 mm thick titanium alloy plate;(3)Rayleigh wave simulation of 4 mm wavelength in 10 mm thick titanium alloy plate;(4)Rayleigh wave simulation of 5 mm wavelength in 10 mm thick titanium alloy plate;(5)A0 mode Lamb Wave simulation of 5 mm wavelength in 4 mm thick titanium alloy plate.

Figure 2c–g show the simulation results, showing the signal amplitude changes of Rayleigh waves with different wavelengths when propagating underwater relative to the air. Figure 2h shows the attenuation of different wavelengths underwater, and the propagation distance is 58.5 mm. The attenuation of different wavelengths is different, that is, the larger the wavelength is, the smaller the attenuation is. The difference is great, especially the attenuation difference of 2 mm and 3 mm wavelength is the largest. The attenuation is expressed by the following formula:$$\alpha =20lg\left(\frac{V_{1}}{V_{2}}\right)$$
where α is Rayleigh wave attenuation unit; dB, $V_{1}\;$is the amplitude in air, and $V_{2}\;$is the amplitude underwater.

The simulation results show that the underwater attenuation of IDT with different structures is very large, which shows that it has important research value and can provide important design parameters for IDT. For thin plates, the existence of upper and lower surfaces leads to the presence of guided Lamb waves. However, the attenuation of short wavelength Rayleigh waves underwater is too large, which is not conducive to structural damage detection. In this paper, a larger wavelength Lamb wave is selected for the simulation. As shown in Figure 1h, the Lamb wave (wavelength is 5 mm) of A0 mode excited on a 4 mm thick titanium alloy plate has similar attenuation with the Rayleigh wave of 5 mm wavelength, which can solve the problem of large attenuation of short wavelength Rayleigh waves on the thin plate.

## 5. Experiment and Result Analysis

### 5.1. Composition of Test System

In order to verify the above simulation results, the experimental system shown in Figure 3a is established. The experimental system can be divided into the excitation part and the receiving amplification part. The excitation principle is that the voltages of adjacent electrodes of the IDT are equal and the potentials are opposite. The excitation part is provided by the signal generator with an 8-cycle two-channel sine wave signal with a phase difference of 180 degrees [17] (Na et al., 2008). The excitation signal is amplified by two common ground single-channel power amplifiers, which are connected to the electrode strips of the IDT. For the receiving part, the signal of receiving IDT is amplified by the instrument amplifier and OP37 amplification and filtering circuit, and finally collected by the oscilloscope.

### 5.2. Experimental Verification of Simulation Results

Five groups of experiments were carried out on the above simulation results. Figure 3b–f shows the results of five groups of experiments: (1) Rayleigh wave propagation experiment at 2 mm wavelength, with a propagation distance of 65 mm. (2) Rayleigh wave propagation experiment with 3 mm wavelength, with a propagation distance of 66 mm. (3) Rayleigh wave propagation experiment with 4 mm wavelength, with a propagation distance of 65 mm. (4) Rayleigh wave propagation experiment with 5 mm wavelength, with a propagation distance of 8 mm. (5) 5 mm Lamb wave propagation experiment, with a propagation distance of 8 mm. Table 2 shows the results after modifying the simulation model according to the actual sensor distance in the experiment. Table 3 shows the peak and peak values of the waveform in Figure 3b–f. Table 2 and Table 3 show that the simulation results of underwater attenuation of Rayleigh waves with 2–5 mm wavelengths are in line with the actual situation. However, for the underwater attenuation of Lamb wave of 5 mm wavelength, the simulation results are different from the actual situation.

### 5.3. Analysis of Experimental Results

The IDT signals of five groups of underwater propagation data collected in the experiment show two waveforms, which are inconsistent with those in the air. For this reason, it is found from Figure 3g that when IDT excites the surface wave, it not only generates elastic waves in the solid domain, but also generates acoustic waves in the water area, and the Rayleigh wave velocity is greater than the underwater acoustic wave velocity, which is also the reason why two acoustic signals appear in the underwater signal in Figure 3b–f.

There is a deviation between the simulation results of the 5 mm wavelength Lamb wave and the experimental results. The reason may be that the Lamb wave received by RX-IDT overlaps with the underwater acoustic signal. As shown in Figure 3f, the frequency of the underwater acoustic signal received by IDT is different from that of the Lamb wave signal, and the signal in the experiment is amplified by the band-pass filter, which shows that the experimental signal is less affected by the underwater acoustic signal than the simulation signal in Figure 3h. In order to reduce the influence of underwater acoustic signal, this paper selects the peak-to-peak value of the signal at 35 μs in Figure 3h for comparison, because the signal at 35 μs is the peak-to-peak value of Lamb wave signal, and the upper limit of the peak value is close to the absolute value of the lower limit of the signal, indicating that the signal at 35 μs is less affected by underwater acoustic interference. The attenuation value of the simulation result is 13.1 dB, which is close to the experimental result of 12.8 dB. Combined with the first four groups of experiments, it is proved that COMSOL software can predict the underwater attenuation of an IDT-excited Lamb wave. The percentage difference between simulation and experiment is shown in Table 3

## 6. Conclusions

In order to realize the application of PVDF film IDT with soft characteristics in deep submersibles, this paper explores and studies the propagation characteristics of surface waves excited by IDT underwater through simulation and experiment. The research work of this paper has the following findings:

(1) Through simulation and experimental study, the underwater attenuation of surface waves excited by IDT is quantified, which provides design parameters for IDT applied in underwater structural damage detection. Additionally, the experimental results show that the attenuation of A0 mode Lamb wave is slightly greater than that of a Rayleigh wave with the same wavelength, but for the damage detection of thin plates, a Lamb wave with a larger wavelength has advantages over Rayleigh wave (in this paper, it is a 4 mm thick titanium alloy plate).

(2) In this paper, the simulation of IDT in five cases by COMSOL software matches with the experimental results, proving that the two-dimensional acoustic structure coupling model with Equation (5) as the boundary condition can accurately predict the attenuation of IDT underwater, which provides a quick verification method for the design of IDT.

(3) When IDT excites the surface wave, it will stimulate the sound wave in the water area, and the sound wave in the water will be received by RX-IDT, that is, the underwater sound in Figure 3b–f. The problem of underwater acoustic interference can be solved by the difference between the acoustic wave velocity in the water and that in the solid.

## Figures and Tables

**Figure 1 sensors-23-01184-f001:**
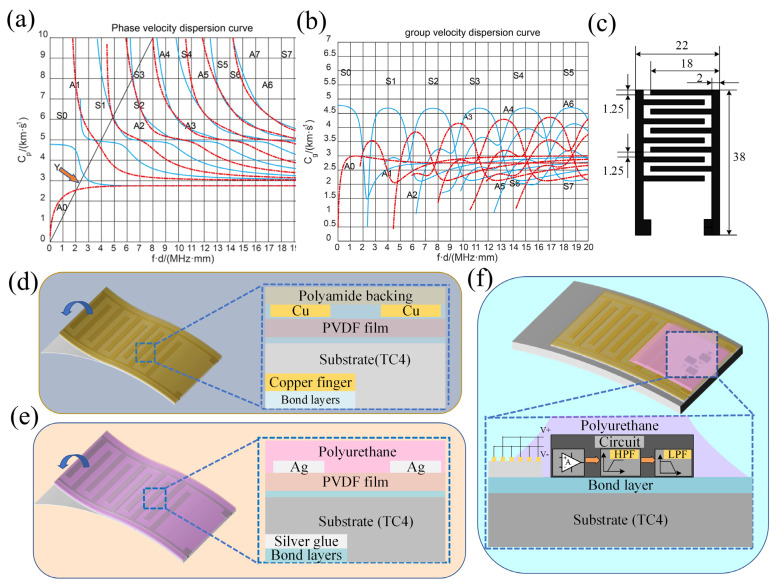
(**a**) Phase Velocity Dispersion Curve of Titanium Alloy Plate. (**b**) Group velocity dispersion curve of titanium alloy plate. (**c**) Electrode pattern parameters of IDT. (**d**) Structure of flexible PCB-IDT. (**e**) Structure of IDT made by screen printing. (**f**) Conceptual diagram of IDT receiving circuit.

**Figure 2 sensors-23-01184-f002:**
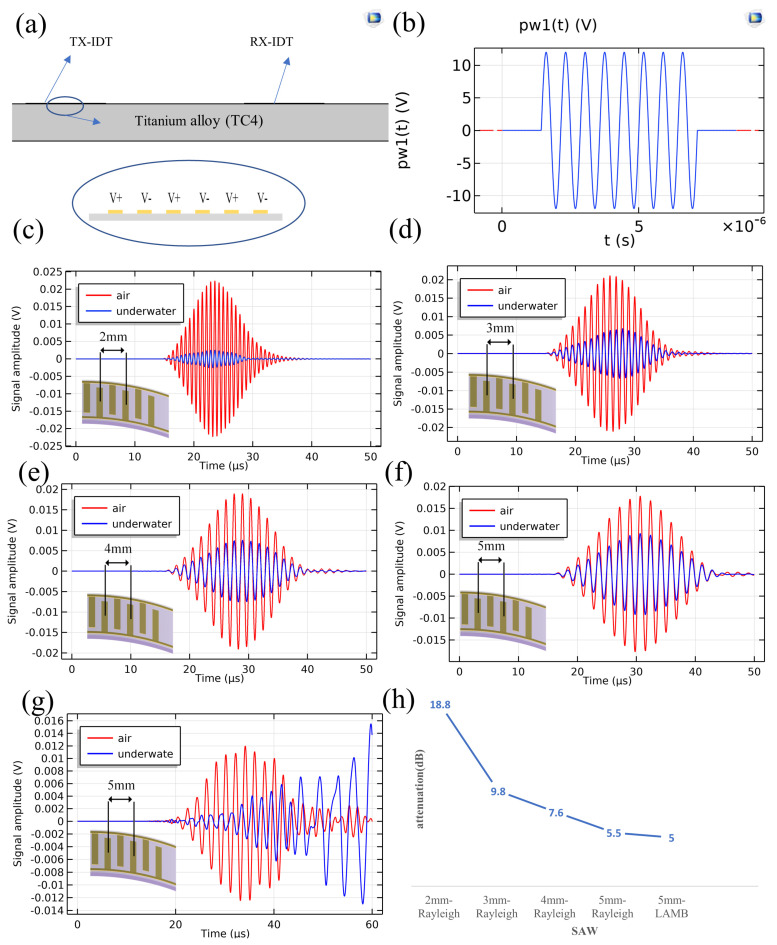
(**a**) Simulation model of IDT. (**b**) IDT excitation signal. (**c**) Simulation results of Rayleigh wave with 2 mm wavelength. (**d**) Simulation results of Rayleigh wave with 3 mm wavelength. (**e**) Simulation results of Rayleigh wave with 4 mm wavelength. (**f**) Simulation results of Rayleigh wave with 5 mm wavelength. (**g**) Simulation results of Lamb wave with 5 mm wavelength. (**h**) Underwater wave attenuation with geometries of transducer.

**Figure 3 sensors-23-01184-f003:**
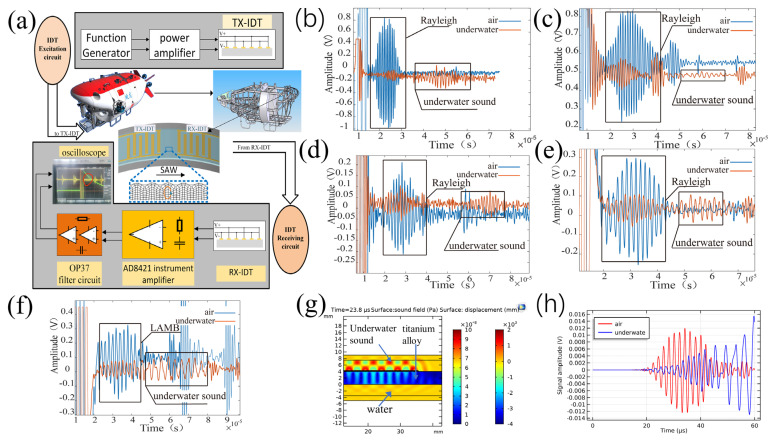
(**a**) Experimental system. (**b**) Experimental signal of Rayleigh wave with 2 mm wavelength. (**c**) Experimental signal of Rayleigh wave with 3 mm wavelength. (**d**) Experimental signal of Rayleigh wave with 4 mm wavelength. (**e**) Experimental signal of Rayleigh wave with 5 mm wavelength. (**f**) Experimental signal of Lamb wave with 5 mm wavelength. (**g**) Underwater acoustic excitation. (**h**) LAMB wave contrast signal with 8 mm spacing.

**Table 1 sensors-23-01184-t001:** Interdigital transducer and titanium alloy parameters.

Wavelength	Interdigital Logarithm	Excitation Frequency	Surface Wave	Interdigital Width
2 mm	11	1.4 MHz	Rayleigh	0.5 mm
3 mm	8	959 kHz	Rayleigh	0.75 mm
4 mm	6	719 kHz	Rayleigh	1 mm
5 mm	5	575 kHz	Rayleigh	1.25 mm
5 mm	5	500 kHz	Lamb	1.25 mm

**Table 2 sensors-23-01184-t002:** COMSOL Simulation Peak to Peak Value and Attenuation.

Wavelength	Surface Wave	Air	Underwater	Attenuation
2 mm	Rayleigh	44.6 mV	3.6 mV	21.8 (dB)
3 mm	Rayleigh	42.2 mV	13 mV	10.2 (dB)
4 mm	Rayleigh	38 mV	15.2 mV	7.96 (dB)
5 mm	Rayleigh	35.8 mV	10.7 mV	10.5 (dB)
5 mm	Lamb	25.4 mV	12 mV	6.5 (dB)

**Table 3 sensors-23-01184-t003:** Peak to peak value and attenuation of experimental data.

Wavelength	Surface Wave	Air	Underwater	Attenuation	Percentage Difference
2 mm	Rayleigh	1.9 V	0.17 V	21 (dB)	4%
3 mm	Rayleigh	0.6 V	0.171 V	10.9 (dB)	7.63%
4 mm	Rayleigh	0.422 V	0.161 V	8.4 (dB)	5.2%
5 mm	Rayleigh	0.551 V	0.167 V	10.4 (dB)	0.96%
5 mm	Lamb	0.496 V	0.113 V	12.8 (dB)	2.34%

## Data Availability

Not applicable.

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
