# Peer review of "Experimental Study of Deep Submersible Structure Defect Monitoring Based on Flexible Interdigital Transducer Surface Acoustic Wave Technology"

_sensors, 2023, doi:10.3390/s23031184_

Round 1

Reviewer 1 Report (Previous Reviewer 3)

Thank you very much for creative consideration of my remarks. Now the work seems to be much better.

Author Response

Reply to comments on manuscript entitled “Experimental Study of Deep Submersible Structure Defect Monitoring based on Flexible interdigital transducer surface acoustic wave technology ”

Thank you for your valuable comments, which gave me a further understanding of the research content of this article. My manuscript has been greatly improved.

Reviewer 2 Report (New Reviewer)

The article is very interesting and raises an important aspect of defect detection. Undoubtedly, the article has scientific and industrial value. The article is suitable for publication in this journal.

However, there are some research questions:

1. What is the percentage difference between a numerical study and an experimental study?

2. Have you tried to use recurrence methods in defect detection in similar research?

Please answer the questions.

Author Response

This manuscript is a resubmission of an earlier submission. The following is a list of the peer review reports and author responses from that submission.

Round 1

Reviewer 1 Report

To solve the shortage of structural defect monitoring methods for deep submersibles, numerical simulation and experimental research on underwater SAW propagation  characteristics based on PVDF-interdigital transducers (PVDF-IDT) of structural health monitoringare carried out in this paper, and the detection of underwater structural defects is realized by exciting surface acoustic waves which can directly detect structural defects.

The typical structure of IDT consists of three layers, the top and bottom electrode layers are separated by piezoelectric layers, and the distance between two adjacent fingers of the interdigital electrode is adjusted to match the specific wavelength.

It is found that LAMB wave has significant advantages over Rayleigh wave in the health monitoring of underwater thin plate structures.

There are some stylistic problems in this paper, eg,

1.Figure stylist, Figure (1) should change to Figure 1,  should describe Figure 5 early and then introduce Figure 6 late in main text , line 148 and line 166.  Figure 10 should change to Figure 9 in line 215.

2. Unit stylist, 500KHz should change to 500kHz in line 107 and Table 1.

3. Reference stylist, capitalize first letter in every word some time, capitalize first letter in first word for title other time. Capitalize first letter in every word some time and capitalize every letter for Journal name other time. The title of reference 15 is in italic type, different other reference.

Reviewer 2 Report

This is a poor-prepared manuscript. All the results about wave velocity and attenuation about Lamb wave in the coupled water-solid media can be analytically analyzed. However, the authors only give the dispersion curve in the single solid structure. F in equation 1 is wrongly explained. Numerous grammar mistakes. The references are too old to reflect the state-of-art. 

Reviewer 3 Report

The paper have to be reconsidered. Many statements are very discussible.

The detailed remarks are given in the attachment.
